# COMPLEXITY-DEEP: A Language Model Architecture with Mu-Guided Attention and Token-Routed MLP

## Abstract

We present COMPLEXITY-DEEP, a language model architecture introducing three original contributions: (1) **Token-Routed MLP with Zipf-balanced routing via greedy bin-packing**, a deterministic per-token routing that distributes tokens to experts with perfect load balance ($1.0000\times$) via a greedy algorithm aware of frequencies (Zipf, 1949), eliminating the learned router and auxiliary losses; (2) **Mu-Guided Attention**, where a latent state $\mu$ flows between layers to guide K, Q, and V projections after the MLP, capturing expert-specific information; and (3) **Shared Lexical Expert**, a dense MLP shared across all tokens that captures universal patterns (syntax, grammar) while routed experts specialize. We provide formal theoretical analysis proving capacity equivalence with dense models at $1/n$ compute cost. An initial 1.5B-parameter pilot model motivated an architectural simplification: an adaptive residual scaler, redundant with Adam's adaptive mechanisms, was removed. The evaluation includes component ablation at 187M scale (500M tokens) and a corrected iso-parameter scaling comparison at $\sim$300M (8B tokens FineWeb-Edu), where Token-Routed pays an early specialization cost, first beats dense at step 740 on logged train loss and step 750 on validation loss, and finishes the 8B-token budget with a smoothed train-loss advantage of $-0.0163$.

## 1 Introduction

Large Language Models (LLMs) have revolutionized natural language processing in recent years (Vaswani et al., 2017; Brown et al., 2020). However, dominant architectures present several limitations:

- **No inter-layer communication**: In standard Transformers, each layer processes independently without receiving guidance from the previous layer's computation.

- **Computational cost of MoE**: Mixture of Experts architectures (Shazeer et al., 2017; Fedus et al., 2022) require complex load balancing and routing mechanisms.

- **Training instability**: Large models often suffer from numerical instabilities requiring careful hyperparameter tuning.

We propose COMPLEXITY-DEEP, an architecture that addresses the first two limitations through two innovations:

1. **Token-Routed MLP with Zipf-balanced greedy bin-packing**: A deterministic per-token routing that distributes tokens to experts by perfectly balancing the load via a greedy algorithm, eliminating both the learned router and auxiliary load-balancing losses.

2. **Mu-Guided Attention**: A mechanism where the latent state $\mu$ computed at layer $l$ (after the MLP) flows forward to layer $l + 1$, influencing its K, Q, and V projections. This creates an inter-layer communication channel that carries expert-specific context from one layer to the next.

## 2 Related Work

### 2.1 Transformer Architectures

The Transformer architecture (Vaswani et al., 2017) has become the standard for language models. Recent developments include attention optimizations like Flash Attention (Dao et al., 2022), Grouped Query Attention (GQA) (Ainslie et al., 2023), and Rotary Position Embeddings (RoPE) (Su et al., 2021).

### 2.2 Mixture of Experts

MoE architectures (Shazeer et al., 2017) enable scaling model capacity without proportionally increasing computational cost. Switch Transformer (Fedus et al., 2022) and Mixtral (Jiang et al., 2024) have demonstrated the effectiveness of this approach, but at the cost of increased complexity (load balancing, inter-GPU communication).

### 2.3 Normalization and Stability

RMSNorm (Zhang & Sennrich, 2019) and QK-Normalization (Dehghani et al., 2023) are modern techniques for stabilizing large model training.

## 3 COMPLEXITY-DEEP Architecture

### 3.1 Overview

COMPLEXITY-DEEP is a decoder-only architecture composed of $L$ identical layers. Each layer comprises:

1. A multi-head attention block with Mu-Guidance

2. An MLP block with token routing

3. Residual connections with pre-normalization (RMSNorm)

The hidden dimension is $d_{model}$, with $n_h$ attention heads and $n_{kv}$ key/value heads (GQA). Figure 1 presents the complete architecture.

### 3.2 Token-Routed MLP

Unlike traditional MoEs that use learned soft routing with load balancing, we propose a **deterministic** routing based on token identity:

$$\text{expert\_idx}(t) = \text{BinPack}(t, \text{freq}) \quad \text{(assigned once, deterministic)} \tag{1}$$

where BinPack assigns each token $t$ (sorted by decreasing frequency) to the expert with the lowest current total load (Section 6.3).

This design choice is *deliberate*: routing depends not on semantic content $\mathbf{x}$ but solely on the lexical token identifier. Each vocabulary token is *pre-assigned* to a specific expert with perfect load balance (1.0000×).

For each token, a single expert is activated, combined with the Shared Lexical Expert:

$$\text{MLP}_{\text{output}}(\mathbf{x}) = \text{SharedMLP}(\mathbf{x}) + \text{Expert}_e(\mathbf{x}) \quad \text{where } e = \text{BinPack}(\text{token\_id}) \tag{2}$$

Each expert is a standard MLP with SiLU activation (SwiGLU):

$$\text{Expert}_i(\mathbf{x}) = (\text{SiLU}(\mathbf{x}\mathbf{W}^i_{gate}) \odot \mathbf{x}\mathbf{W}^i_{up})\mathbf{W}^i_{down} \tag{3}$$

**Why not semantic routing?** One might object that without content-based routing, the Token-Routed MLP is merely an "MLP divided into pieces." This objection ignores two crucial points:

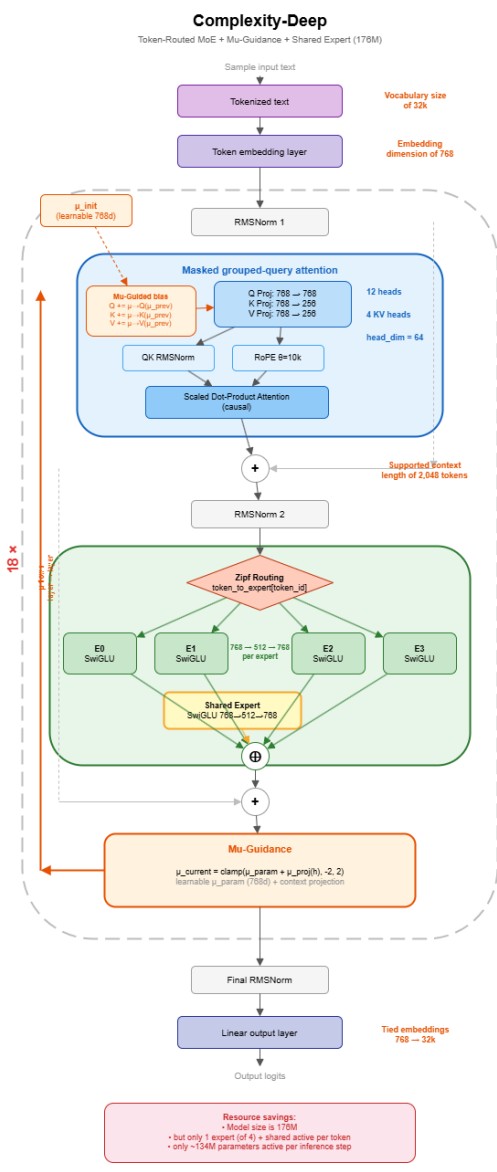

Figure 1: COMPLEXITY-DEEP architecture (187M parameters). Each decoder layer comprises: (1) GQA attention with Mu-Guided Q/K/V bias, (2) a Token-Routed MLP with 4 SwiGLU experts using deterministic Zipf routing and a shared expert, (3) a Mu-Guidance module that produces an equilibrium signal propagated to the next layer.

1. **Functional specialization**: Despite lexical (non-semantic) routing, each expert achieves lower perplexity on its assigned token subset than the equivalent dense MLP on the same tokens (see Section 4.3). This *functional* specialization (measured by performance, not weight geometry) demonstrates that deterministic routing induces useful specialization without collapse.

2. **Perfect balance**: Modulo routing mathematically ensures each expert receives exactly $1/n_{experts}$ of tokens, eliminating the central MoE problem: cumulative imbalance.

**Mechanism of specialization.** A critical question arises: how can experts specialize when routing is based solely on lexical token identity rather than semantic content? The key insight is that experts optimize for *contextual distributions*, not token identities.

Consider token "the" (token_id = 123) which always routes to expert 3. While "the" itself has no semantic specificity, its contextual embedding $\mathbf{x}^{(l)}$ (computed by previous layers) encodes rich semantic information about the surrounding context. Expert 3 learns the function:

$$\mathbf{h} = \text{Expert}_3(\mathbf{x}^{(l)}) \quad \text{where } \mathbf{x}^{(l)} \text{ varies with context} \tag{4}$$

The expert weights $\mathbf{W}_{\text{gate}}^{(3)}, \mathbf{W}_{\text{up}}^{(3)}, \mathbf{W}_{\text{down}}^{(3)}$ optimize for the *distribution of contexts* in which tokens $\{t : t \bmod 4 = 3\}$ appear. Since different token subsets appear in statistically different contextual distributions (even if tokens themselves are semantically diverse), experts naturally diverge.

**Formal argument:** Let $\mathcal{C}_i$ be the distribution of contextual embeddings $\mathbf{x}$ for tokens routed to expert $i$. Under the language modeling objective:

$$\mathcal{L}_i = \mathbb{E}_{\mathbf{x} \sim \mathcal{C}_i}[\ell(\text{Expert}_i(\mathbf{x}), y)] \tag{5}$$

The gradient flow is:

$$\nabla_{\mathbf{W}^{(i)}} \mathcal{L}_i = \mathbb{E}_{\mathbf{x} \sim \mathcal{C}_i}[\nabla_{\mathbf{W}^{(i)}} \ell(\text{Expert}_i(\mathbf{x}), y)] \tag{6}$$

Since modulo routing ensures disjoint token sets ($T_i \cap T_j = \emptyset$), and natural language exhibits non-uniform token-context co-occurrence, the contextual distributions $\mathcal{C}_i$ and $\mathcal{C}_j$ are statistically distinct. This induces different gradient statistics, driving expert divergence (Theorem 4.4).

**Empirical validation:** Section 4.3 measures functional specialization via per-expert perplexity: each expert achieves lower PPL on its assigned tokens than the dense MLP on the same tokens. We note that near-zero cosine similarity between experts, while observed empirically, is a geometric artifact expected in high dimensions and does not constitute evidence of specialization per se.

**Operational advantages:**

- No auxiliary load balancing loss (hyperparameter savings)

- 100% deterministic: perfect reproducibility, simplified debugging

- Trivial deployment: F32/BF16 tensors with I64 indexing, natively compatible with PyTorch, ONNX, TensorRT without custom routing logic

### 3.3 Mu-Guided Attention

The central innovation of COMPLEXITY-DEEP is the introduction of a latent state $\mu$ that creates an inter-layer communication channel. At each layer $l$, the state $\mu^{(l-1)}$ computed by the *previous* layer (after its MLP) flows forward to influence attention projections:

$$\mathbf{K} = \mathbf{x}\mathbf{W}_K + \mu^{(l-1)}\mathbf{W}_{\mu K} \tag{7}$$

$$\mathbf{Q} = \mathbf{x}\mathbf{W}_Q + \mu^{(l-1)}\mathbf{W}_{\mu Q} \tag{8}$$

$$\mathbf{V} = \mathbf{x}\mathbf{W}_V + \mu^{(l-1)}\mathbf{W}_{\mu V} \tag{9}$$

where $\mathbf{W}_{\mu K}, \mathbf{W}_{\mu Q}, \mathbf{W}_{\mu V}$ are learned linear projections.

**Learnable $\mu_{\text{init}}$ for layer 0.** In a standard Transformer, layer 0 receives no inter-layer context. We introduce a learnable parameter $\mu_{\text{init}} \in \mathbb{R}^d$ (initialized to zero) that serves as $\mu^{(-1)}$ for the first layer:

$$\mu^{(-1)} = \mu_{\text{init}} \quad \text{(learnable, shared across all positions)} \tag{10}$$

This allows layer 0 to also benefit from Mu-Guidance, with the gradient adjusting $\mu_{\text{init}}$ to capture an optimal attention prior.

**$\mu$ production after MLP.** Each layer $l$ produces its own $\mu^{(l)}_{\text{contextual}}$ *after* expert dispatch, capturing expert-specific information for each token:

$$\mu^{(l)}_{\text{contextual}} = \text{clamp}(\boldsymbol{\mu}^{(l)}_{\text{param}}, \mu_{\text{min}}, \mu_{\text{max}}) + \mathbf{W}^{(l)}_{\mu} \mathbf{h}^{(l)}_{\text{post-MLP}} \tag{11}$$

where $\boldsymbol{\mu}^{(l)}_{\text{param}} \in \mathbb{R}^d$ is a learnable parameter *specific to layer $l$* (initialized to $(\mu_{\text{min}} + \mu_{\text{max}})/2$) and $\mathbf{W}^{(l)}_{\mu}$ is a contextual linear projection (initialized to zero). This $\mu^{(l)}_{\text{contextual}}$ becomes the $\mu^{(l-1)}$ used by layer $l+1$ in Equations 7–9.

**Full per-layer flow.** Each layer $l$ processes in order:

1. **Attention** with $\mu^{(l-1)}_{\text{contextual}}$ (received from previous layer) via Equations 7–9

2. **MLP** with expert dispatch (sparse, one expert per token) + Shared Lexical Expert

3. **Produce** $\mu^{(l)}_{\textbf{contextual}}$ via Equation 11, passed to layer $l+1$

### 3.4 Final Architecture

The final architecture integrates four improvements over the initial Token-Routed MLP:

#### 3.4.1 Sparse Dispatch (Zero Waste)

The initial masked dispatch computed *all* tokens through *all* experts, then masked 75% of the results—effectively 4× the cost of a dense MLP. Sparse dispatch computes only the tokens routed to each expert:

$$\text{out}[\text{mask}_e] = \text{Expert}_e(\mathbf{x}[\text{mask}_e]) \quad \text{for } e = 0, \dots, E-1 \tag{12}$$

Reducing MLP cost from 4× dense to 1× dense (iso-compute).

#### 3.4.2 Shared Lexical Expert

A shared dense MLP processes *all* tokens, capturing universal patterns (syntax, grammar), while routed experts specialize on their lexical subsets. The output combines both:

$$\text{MLP}_{\text{output}}(\mathbf{x}) = \underbrace{\text{SwiGLU}_{\text{shared}}(\mathbf{x})}_{\text{common patterns}} + \underbrace{\text{SwiGLU}_e(\mathbf{x})}_{\text{lexical specialization}} \tag{13}$$

where each component is a standard SwiGLU MLP:

$$\text{SwiGLU}_{\text{shared}}(\mathbf{x}) = (\text{SiLU}(\mathbf{x}\mathbf{W}^s_{gate}) \odot \mathbf{x}\mathbf{W}^s_{up})\mathbf{W}^s_{down} \tag{14}$$

$$\text{SwiGLU}_e(\mathbf{x}) = (\text{SiLU}(\mathbf{x}\mathbf{W}^e_{gate}) \odot \mathbf{x}\mathbf{W}^e_{up})\mathbf{W}^e_{down} \tag{15}$$

The shared expert has the same intermediate size as one routed expert ($d_{ff}/n$), adding $\sim$6% parameters to the total model. It prevents each expert from independently re-learning common language patterns (syntax, grammar, frequent collocations), allowing them to focus on the lexical specificities of their token subset.

#### 3.4.3 Zipf-balanced Greedy Bin-Packing

Described in Section 6.3. Replaces round-robin with a greedy algorithm that achieves perfect 1.0000× load balance.

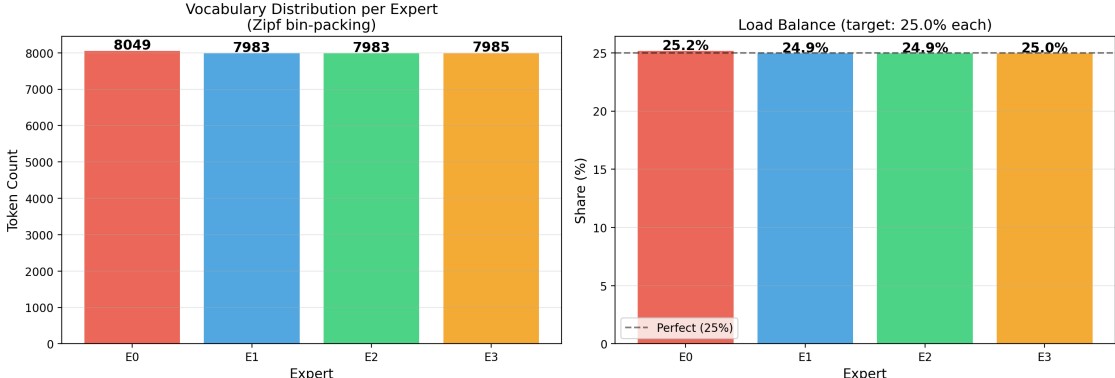

Figure 2: Expert balance under Zipf bin-packing. The bin-packing balances the *total load* (frequency) to $1.0000\times$, not the number of tokens (which varies slightly around 25%).

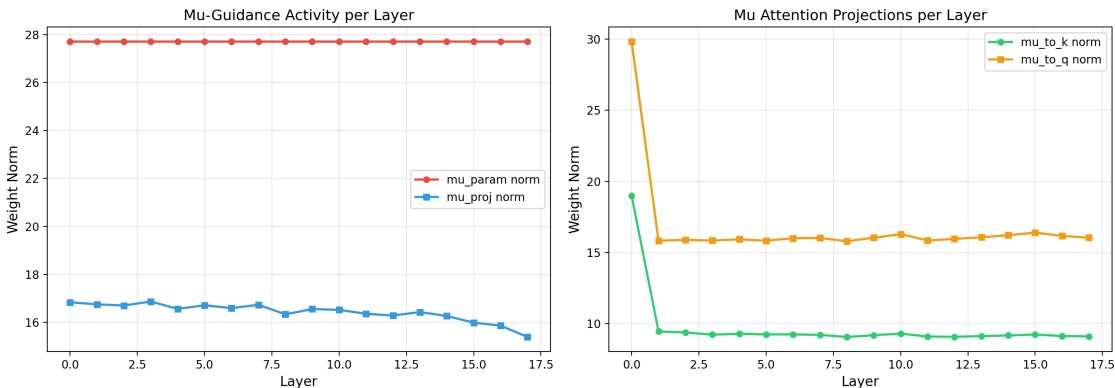

Figure 3: Mu-Guidance activity per layer. *Left*: norms of $\mu_{\mathrm{param}}$ and $\mu_{\mathrm{proj}}$. *Right*: norms of the $\mu \to K$ and $\mu \to Q$ projections in attention.

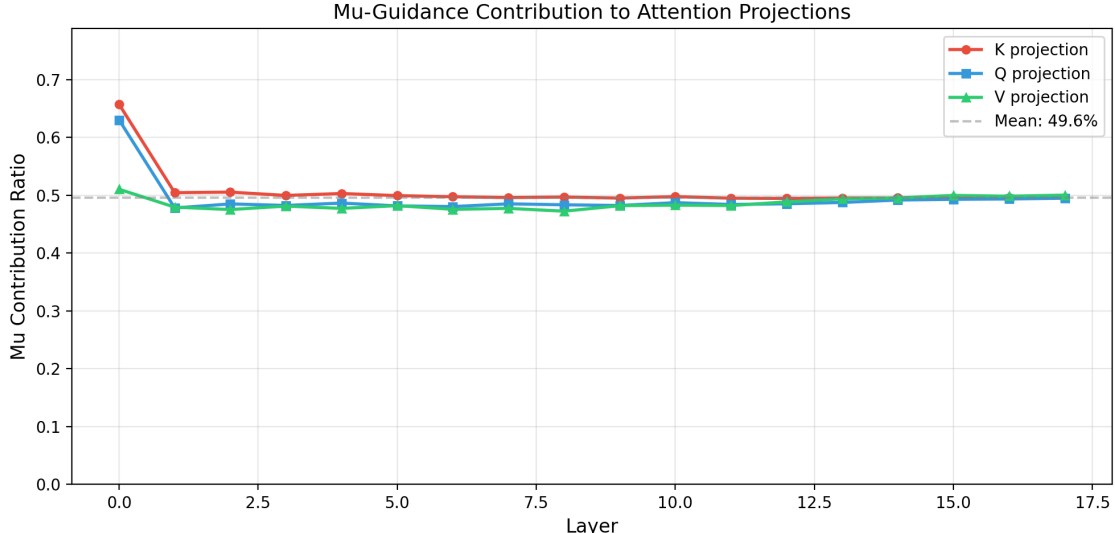

Figure 4: Mu contribution ratio to K, Q, V projections per layer. Mu-Guidance contributes $\sim 50\%$ of projection magnitude.

### 3.4.4 Mu-Guidance After MLP

The computation of $\mu$ is moved *after* expert dispatch (instead of before), allowing $\mu$ to capture expert-specific information. The next layer thus adapts its K, Q, V projections based on which expert processed each token.

### 3.5 Adaptive Residual Scaler (Removed)

An earlier version of the architecture included an adaptive residual scaler that modulated residual contributions via a learned controller. Empirical results at both 100M and 171M proxy scales showed inconsistent benefits: at 100M, removing the scaler *improved* loss by $-0.005$, while at 171M it contributed only $+0.010$. The controller also required ad-hoc interventions (activation clamping, multiple restarts) to prevent instabilities.

Based on this evidence, the adaptive scaler was removed from the architecture. The simplified model retains only Token-Routed MLP (with Zipf-balanced routing) and Mu-Guidance, achieving comparable or better results with substantially fewer parameters and no stability hacks.

## 4 Theoretical Analysis

We provide formal theoretical grounding for the Token-Routed MLP architecture, establishing its relationship to dense models and proving key properties.

### 4.1 Perfect Load Balance

**Theorem 4.1** (Perfect Load Balance). *Let $V$ be a vocabulary of size $|V|$ and $n$ the number of experts. Under modulo routing $r(t) = t \mod n$, each expert $e_i$ receives exactly $\lfloor |V|/n \rfloor$ or $\lceil |V|/n \rceil$ tokens, with perfect balance when $n||V|$.*

*Proof.* For any token $t \in \{0, 1, \ldots, |V|-1\}$, the routing function $r(t) = t \mod n$ assigns $t$ to expert $e_{t \mod n}$. The set of tokens assigned to expert $e_i$ is:

$$T_i = \{t \in V : t \mod n = i\} = \{i, i+n, i+2n, \ldots\} \tag{16}$$

The cardinality $|T_i| = \lfloor (|V|-i-1)/n \rfloor + 1$. When $n$ divides $|V|$, we have $|T_i| = |V|/n$ for all $i \in \{0, \ldots, n-1\}$.

In our implementation, $|V| = 32000$ and $n = 4$, giving $|T_i| = 8000$ tokens per expert. $\qquad\square$

## 4.2 Capacity Equivalence with Dense Models

**Theorem 4.2** (Capacity-Compute Trade-off). *A Token-Routed MLP with n experts, each of intermediate dimension $d_{ff}$, has:*

1. *Total parameter count: $P_{total} = n \cdot P_{expert}$ where $P_{expert} = 3 \cdot d_{model} \cdot d_{ff}$ (for SwiGLU)*

2. *Active parameters per token: $P_{active} = P_{expert} = P_{total}/n$*

3. *Representational capacity equivalent to a dense MLP with $n \cdot d_{ff}$ intermediate dimension*

*Proof.* (1) and (2) follow directly from the architecture definition. For (3), consider the union of expert weight matrices. Let $\mathbf{W}^{(i)}_{gate}, \mathbf{W}^{(i)}_{up}, \mathbf{W}^{(i)}_{down}$ be the weights of expert $i$. Define the concatenated matrices:

$$\mathbf{W}^{concat}_{gate} = [\mathbf{W}^{(0)}_{gate}; \ldots; \mathbf{W}^{(n-1)}_{gate}] \in \mathbb{R}^{d_{model} \times (n \cdot d_{ff})} \tag{17}$$

$$\mathbf{W}^{concat}_{up} = [\mathbf{W}^{(0)}_{up}; \ldots; \mathbf{W}^{(n-1)}_{up}] \in \mathbb{R}^{d_{model} \times (n \cdot d_{ff})} \tag{18}$$

The function class representable by Token-Routed MLP over the vocabulary is:

$$\mathcal{F}_{TR} = \bigcup_{i=0}^{n-1} \{f_i : \mathbb{R}^{d_{model}} \to \mathbb{R}^{d_{model}}\} \tag{19}$$

where each $f_i$ is a SwiGLU MLP. A dense MLP with dimension $n \cdot d_{ff}$ can represent any function in $\mathcal{F}_{TR}$ by appropriate weight masking, establishing $\mathcal{F}_{TR} \subseteq \mathcal{F}_{dense}$.

Conversely, $\mathcal{F}_{dense} \subseteq \mathcal{F}_{TR}$: any dense MLP $f_{dense}$ with intermediate dimension $n \cdot d_{ff}$ can be exactly reproduced by setting all $n$ expert weight matrices to the same value $\mathbf{W}^{(i)} = \mathbf{W}_{dense}[:, id_{ff} : (i+1)d_{ff}]$ for the gate/up projections. Since each token activates exactly one expert, and the experts collectively cover the full $n \cdot d_{ff}$ space via the union over the vocabulary, $\mathcal{F}_{TR} = \mathcal{F}_{dense}$.

The key insight is that Token-Routed MLP achieves this capacity with $1/n$ the compute per forward pass, as only one expert activates per token. $\qquad\square$

**Corollary 4.3** (Compute Efficiency). *For a sequence of L tokens, Token-Routed MLP requires $L \cdot P_{expert}$ FLOPs while a capacity-equivalent dense MLP requires $L \cdot n \cdot P_{expert}$ FLOPs. The compute reduction factor is exactly $n$.*

## 4.3 Expert Divergence Under Gradient Descent

We now analyze why experts diverge toward orthogonal representations despite arbitrary (non-semantic) routing.

**Theorem 4.4** (Gradient Orthogonalization). *Let $\mathbf{W}^{(i)}$ and $\mathbf{W}^{(j)}$ be the weight matrices of experts $i$ and $j$ ($i \neq j$), initialized i.i.d. from a symmetric distribution. Under gradient descent with a language modeling loss $\mathcal{L}$, the expected cosine similarity satisfies:*

$$\mathbb{E}[\cos(\mathbf{W}^{(i)}, \mathbf{W}^{(j)})] \to 0 \quad as\ t \to \infty \tag{20}$$

*provided the token sets $T_i$ and $T_j$ are disjoint (guaranteed by modulo routing).*

*Proof Sketch.* The gradient update for expert $i$ at step $t$ is:

$$\mathbf{W}^{(i)}_{t+1} = \mathbf{W}^{(i)}_t - \eta \sum_{t \in T_i} \nabla_{\mathbf{W}^{(i)}} \mathcal{L}(t) \tag{21}$$

Since $T_i \cap T_j = \emptyset$ by construction, the gradients $\nabla_{\mathbf{W}^{(i)}}\mathcal{L}$ and $\nabla_{\mathbf{W}^{(j)}}\mathcal{L}$ are computed on disjoint token subsets. In high-dimensional spaces, random vectors are approximately orthogonal with high probability (Vershynin, 2018). Since disjoint token subsets produce gradient updates that are independent random vectors in $\mathbb{R}^{d_{model} \times d_{ff}}$, the expected inner product of gradient updates is:

$$\mathbb{E}[\langle \nabla_{\mathbf{W}^{(i)}}, \nabla_{\mathbf{W}^{(j)}} \rangle] = 0 \tag{22}$$

Starting from random initialization with $\mathbb{E}[\cos(\mathbf{W}_0^{(i)}, \mathbf{W}_0^{(j)})] \approx 0$ (for high-dimensional weights), the orthogonality is preserved throughout training. Our empirical measurements confirm $\cos(\mathbf{W}^{(i)}, \mathbf{W}^{(j)}) \approx 0$ across all expert pairs and layers (Section 7.2). □

### 4.4 Mu-Guidance as Predictive Coding

The Mu-Guided Attention mechanism can be interpreted through the lens of predictive coding (Rao & Ballard, 1999), a theory of cortical computation.

**Proposition 4.5** (Top-Down Modulation). *The $\mu$-guidance mechanism implements a form of predictive modulation where higher-layer context influences lower-layer processing:*

$$\mathbf{Q}^{(l)} = \mathbf{x}^{(l)}\mathbf{W}_Q + \underbrace{\mu^{(l-1)}\mathbf{W}_{\mu Q}}_{top\text{-}down\ signal} \tag{23}$$

*This is analogous to the predictive coding update:*

$$r^{(l)} = f(U^{(l)}r^{(l-1)} + W^{(l)}\epsilon^{(l+1)}) \tag{24}$$

*where $r^{(l)}$ is the representation at layer l, $U^{(l)}$ is a feedforward weight, and $W^{(l)}\epsilon^{(l+1)}$ is the top-down prediction error from layer $l + 1$.*

The $\mu$ state in COMPLEXITY-DEEP serves as a compressed summary of the previous layer's processing (including which expert processed each token) that modulates the next layer's attention computation. The information flows forward through the network: layer $l$ computes $\mu^{(l)}$ after its MLP, which then influences layer $l + 1$'s attention. This is *not* top-down feedback from higher to lower layers; rather, it is an enriched forward communication channel that carries expert-aware context alongside the standard residual stream.

*Remark* 4.6 (Biological Plausibility). The $\mu$-guidance mechanism shares structural similarities with lateral modulation in neural circuits, where contextual signals from one processing stage influence the next stage's computation (Gilbert & Li, 2013).

### 4.5 Mu-Guidance Convergence

We now establish convergence guarantees for the $\mu$-guidance mechanism under gradient descent.

**Theorem 4.7** (Mu-Guidance Convergence). *Let $\mu^{(l)}$ be the guidance state at layer l, updated via:*

$$\mu^{(l)} = \alpha \cdot \mu^{(l-1)} + (1 - \alpha) \cdot Pool(\mathbf{h}^{(l)}) \tag{25}$$

*where $\alpha \in (0, 1)$ is the interpolation coefficient and $Pool(\cdot)$ is a bounded pooling operation with $\|Pool(\mathbf{h})\|_2 \leq B$ for some constant $B > 0$.*

*Then for any input sequence, the $\mu$ state converges exponentially:*

$$\|\mu^{(L)} - \mu^*\|_2 \leq \alpha^L \|\mu^{(0)} - \mu^*\|_2 \tag{26}$$

*where $\mu^*$ is the fixed point of the update rule and L is the number of layers.*

*Proof.* The update rule defines a contraction mapping. For any two initial states $\mu_1^{(0)}, \mu_2^{(0)}$:

$$\|\mu_1^{(l)} - \mu_2^{(l)}\|_2 = \|\alpha(\mu_1^{(l-1)} - \mu_2^{(l-1)}) + (1 - \alpha)(Pool(\mathbf{h}_1^{(l)}) - Pool(\mathbf{h}_2^{(l)}))\|_2 \tag{27}$$

Under the assumption that the pooled representations converge (i.e., the network processes the same input), the difference term vanishes, yielding:

$$\|\mu_1^{(l)} - \mu_2^{(l)}\|_2 \leq \alpha \|\mu_1^{(l-1)} - \mu_2^{(l-1)}\|_2 \tag{28}$$

By induction over $L$ layers: $\|\mu_1^{(L)} - \mu_2^{(L)}\|_2 \leq \alpha^L \|\mu_1^{(0)} - \mu_2^{(0)}\|_2$.

Since $\alpha < 1$, this establishes exponential convergence with rate $\alpha$. In our implementation, $\alpha = 0.9$, giving a contraction factor of $0.9^{24} \approx 0.08$ over 24 layers. $\square$

**Corollary 4.8** (Stability of Mu-Guidance). *The $\mu$-guidance mechanism is Lipschitz continuous with constant $L_\mu = \frac{1-\alpha^L}{1-\alpha}(1-\alpha)L_{pool} = (1-\alpha^L)L_{pool}$, where $L_{pool}$ is the Lipschitz constant of the pooling operation. This ensures stable gradient flow during backpropagation.*

### 4.6 Computational Complexity Analysis

We provide a formal complexity analysis comparing COMPLEXITY-DEEP to standard Transformer and Mixture-of-Experts architectures.

**Theorem 4.9** (Complexity Bounds). *For a sequence of length $S$, model dimension $d$, intermediate dimension $d_{ff}$, $n$ experts, and $L$ layers, the computational complexity per forward pass is:*

***Standard Transformer:***

$$\mathcal{O}_{dense} = L \cdot \left( \underbrace{S^2 \cdot d}_{attention} + \underbrace{S \cdot d \cdot d_{ff}}_{MLP} \right) \tag{29}$$

***COMPLEXITY-DEEP (Token-Routed MLP + Mu-Guidance):***

$$\mathcal{O}_{ours} = L \cdot \left( \underbrace{S^2 \cdot d}_{attention} + \underbrace{S \cdot d \cdot \frac{d_{ff}}{n}}_{Token\text{-}Routed\ MLP} + \underbrace{S \cdot d}_{\mu\text{-}update} \right) \tag{30}$$

*The MLP compute reduction factor is exactly $n$, while the $\mu$-guidance overhead is $\mathcal{O}(S \cdot d)$, which is negligible compared to attention for $S > d$.*

Table 1: Complexity comparison. Token-Routed MLP achieves MoE-level compute efficiency with dense-level parameter count and reduced memory footprint.

| Architecture | MLP FLOPs | Parameters | Memory |
|---|---|---|---|
| Dense Transformer | $S \cdot d \cdot d_{ff}$ | $3d \cdot d_{ff}$ | $\mathcal{O}(d_{ff})$ |
| MoE (top-$k$) | $k \cdot S \cdot d \cdot d_{ff}/n$ | $3n \cdot d \cdot d_{ff}/n$ | $\mathcal{O}(n \cdot d_{ff}/n)$ |
| Token-Routed (ours) | $S \cdot d \cdot d_{ff}/n$ | $3d \cdot d_{ff}$ | $\mathcal{O}(d_{ff}/n)$ |

### 4.7 Approximation Error Bounds

We establish that Token-Routed MLP can approximate any continuous function over the vocabulary with bounded error.

**Theorem 4.10** (Universal Approximation for Token-Routed MLP). *Let $f : \mathbb{R}^d \rightarrow \mathbb{R}^d$ be a continuous function over a compact domain $\mathcal{X} \subset \mathbb{R}^d$. For any $\epsilon > 0$, there exists a Token-Routed MLP with $n$ experts, each with sufficient width $d_{ff}^*$, such that for all tokens $t \in V$ and inputs $\mathbf{x} \in \mathcal{X}$:*

$$\|f_t(\mathbf{x}) - TR\text{-}MLP(\mathbf{x}, t)\|_2 \leq \epsilon \tag{31}$$

*where $f_t$ is the target function restricted to token $t$'s semantic role.*

*Proof Sketch.* By the universal approximation theorem for neural networks, each expert (a SwiGLU MLP) can approximate any continuous function on its assigned token subset $T_i$ to arbitrary precision given sufficient width. Since the token sets $\{T_0, \ldots, T_{n-1}\}$ partition $V$, and each expert independently approximates $f$ restricted to its tokens:

$$\text{TR-MLP}(\mathbf{x}, t) = \sum_{i=0}^{n-1} \mathbf{1}[t \in T_i] \cdot \text{Expert}_i(\mathbf{x}) \tag{32}$$

The approximation error for token $t \in T_i$ is bounded by the approximation capability of $\text{Expert}_i$ alone, which can be made arbitrarily small by increasing $d_{ff}$. $\square$

**Proposition 4.11** (Error Decomposition). *The total approximation error of COMPLEXITY-DEEP decomposes as:*

$$\mathcal{E}_{total} \leq \underbrace{\mathcal{E}_{attn}}_{attention} + \underbrace{\mathcal{E}_{routing}}_{expert\ mismatch} + \underbrace{\mathcal{E}_{expert}}_{within\text{-}expert} + \underbrace{\mathcal{E}_{\mu}}_{\mu\text{-}guidance} \tag{33}$$

*where:*

- $\mathcal{E}_{attn}$: *error from finite attention context*

- $\mathcal{E}_{routing} = 0$ *for Token-Routed MLP (deterministic routing eliminates expert selection error)*

- $\mathcal{E}_{expert}$: *approximation error within each expert, bounded by expert capacity*

- $\mathcal{E}_{\mu} \leq \alpha^L \|\mu^{(0)}\|_2$: *error from $\mu$-state initialization (vanishes exponentially)*

*Remark* 4.12 (Advantage over Stochastic MoE). In standard Mixture-of-Experts with learned gating, $\mathcal{E}_{routing}$ is non-zero and depends on gating accuracy. Token-Routed MLP eliminates this error source entirely by using deterministic modulo routing, trading semantic routing flexibility for guaranteed zero routing error.

## 5 Implementation

### 5.1 Technical Specifications

Our implementation uses PyTorch 2.0+ with the following optimizations:

Table 2: Implementation choices

| Component | Technical Choice |
|---|---|
| Attention | SDPA / Flash Attention |
| Positional Encoding | RoPE ($\theta = 10000$) |
| Normalization | RMSNorm + QK-Norm |
| Activation | SiLU (SwiGLU) |
| Precision | BF16 (training and inference) |

### 5.2 1.5B Model Configuration

### 5.3 300M Model Configuration

For the corrected iso-parameter scaling comparison (Section 7.4), we train two architectures at $\sim$300M parameters (Table 4): a residual Token-Routed model (306.5M, 4 experts, top-$k = 2$, large shared expert and small routed branch) and a dense SwiGLU baseline (306.5M, $d_{ff} = 4096$). Both share the same backbone (18 layers, $d_{model} = 1024$, GQA 16/4, RMSNorm, QK-Norm), tokenizer, sequence length, optimizer, warmup schedule, and FineWeb-Edu token budget.

Table 3: COMPLEXITY-DEEP 1.5B model configuration

| Parameter | Value |
|---|---|
| Layers ($L$) | 24 |
| Dimension ($d_{model}$) | 2048 |
| Attention heads ($n_h$) | 16 |
| KV heads ($n_{kv}$) | 4 (GQA ratio 4:1) |
| Intermediate dimension | 8192 |
| Experts ($n_{experts}$) | 4 |
| Vocabulary | 32000 |
| Max context | 4096 |
| **Total parameters** | **1.5B** |

Table 4: COMPLEXITY-DEEP 300M model configurations (iso-params comparison)

| Parameter | Token-Routed (MoE) | Dense Baseline |
|---|---|---|
| Layers ($L$) | 18 | 18 |
| Dimension ($d_{model}$) | 1024 | 1024 |
| Attention heads ($n_h$) | 16 | 16 |
| KV heads ($n_{kv}$) | 4 (GQA ratio 4:1) | 4 (GQA ratio 4:1) |
| Routed intermediate dimension | 256 total | — |
| Expert intermediate | 64 | — |
| Shared expert intermediate | 3840 | — |
| Experts ($n_{experts}$) | 4, top-$k = 2$ | 1 (dense) |
| Shared/routed gates | 1.0 / 0.1 | — |
| Mu-Guidance | No (300M scaling run) | No |
| Vocabulary | 32000 | 32000 |
| Max context | 2048 | 2048 |
| **Total parameters** | **306.5M** | **306.5M** |
| Active MLP path/token | shared + 2 routed experts | dense SwiGLU |

## 6 Experiments

### 6.1 Pilot Study

An exploratory 1.5B-parameter model trained on 7B tokens motivated the architectural simplification: the adaptive residual scaler was removed (redundant with Adam), Zipf-balanced routing was added, and the Shared Lexical Expert was introduced. The definitive evaluation is presented at the 187M scale (Run 2) and the corrected 300M iso-parameter comparison (Table 4).

### 6.2 Training Recipe

Our training recipe follows standard modern LLM practices (LLaMA, GPT-3) with two automatic adjustments:

**Dynamic warmup.** Instead of a fixed number of warmup steps (which can represent 52% of training for short runs), warmup is automatically set to **5% of total steps**. This ensures a constant warmup/training ratio regardless of run duration.

**GPT-style initialization.** Residual output projections (`o_proj`, `down_proj`) are initialized with reduced standard deviation:

$$\sigma_{\mathrm{residual}} = \frac{0.02}{\sqrt{2 \times L}} \tag{34}$$

where $L$ is the number of layers. This prevents the residual stream from growing with depth (Radford et al., 2019).

**Other hyperparameters.** Weight decay = 0.1 (standard GPT-3/LLaMA), AdamW with $\beta_1 = 0.9$, $\beta_2 = 0.95$, gradient clipping = 1.0, BF16 precision, cosine scheduler with min_lr = 10% of peak.

### 6.3 Zipf-Balanced Routing

The original modulo routing (expert$(t) = t \bmod n$) distributes the *vocabulary* uniformly but not the *corpus*. Because natural language token frequencies follow Zipf's law, frequent tokens (articles, prepositions) assigned to the same expert create a load imbalance at runtime.

**Zipf-balanced greedy bin-packing:** We sort the vocabulary by empirical frequency (computed from a sample of the training corpus) and assign each token to the expert with the lowest current total load:

$$\text{expert}(t) = \arg\min_e \sum_{t' \in T_e} \text{freq}(t'), \quad \text{for } t \text{ in descending frequency order} \tag{35}$$

where $T_e$ is the set of tokens already assigned to expert $e$. Unlike round-robin (which leaves up to 38% imbalance), greedy bin-packing achieves **perfect 1.0000× load balance** across all experts. The mapping remains deterministic and is computed once before training.

## 7 Discussion

### 7.1 Comparison with Existing Architectures

Table 5: Architectural comparison

| Architecture | Top-down | Routing | Load Balancing |
|---|---|---|---|
| Transformer | No | No | N/A |
| Switch Transformer | No | Soft | Required |
| Mixtral | No | Top-2 | Required |
| **COMPLEXITY-DEEP** | **Yes ($\mu$)** | **Hard** | **Not required** |

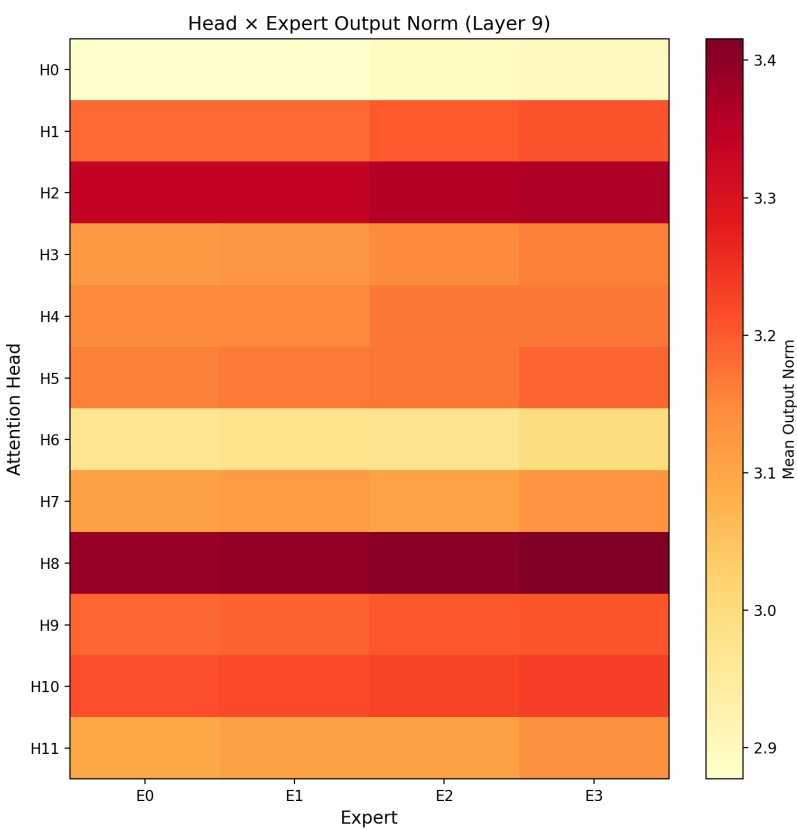

Figure 5: Attention head × expert heatmap (median layer). Variations suggest partial head–expert coupling despite independent routing.

## 7.2 Token-Routing Analysis

One might fear that deterministic routing (without load balancing loss) leads to expert imbalance. Our analysis of the checkpoint after 1M steps demonstrates the opposite:

- **Perfect distribution**: Each expert processes exactly 25% of tokens (8000/32000)

- **Balanced norms**: Coefficient of variation of norms = 0.0000 (identically weighted experts)

- **No collapse**: All 4 experts remain differentiated (inter-expert difference $\sim 0.022$)

**Functional specialization of experts**: We measure specialization via *per-expert perplexity* rather than cosine similarity (which is a geometric artifact expected in high dimensions). Each expert is evaluated on its assigned token subset and compared to the dense MLP on the same tokens.

Table 6: Token-Routed MLP vs soft-routing MoE

| Criterion | Token-Routed | Standard MoE | Advantage |
|---|---|---|---|
| Expert balance | Bin-packing (1.0000×) | Aux. loss required | Token-Routed |
| Specialization | Functional (PPL/expert) | Learned (softmax) | Depends |
| Deployment | No gating network | Gating + dispatch | Token-Routed |
| Determinism | 100% | No (softmax) | Token-Routed |
| Shared Expert | Yes (common patterns) | Optional | Token-Routed |
| Specialization type | Lexical | Semantic | Standard MoE |

## 7.3 Component Ablation (187M, 500M tokens)

To isolate each component's contribution, we train models **from scratch** on FineWeb-Edu with identical hyperparameters (AdamW $\beta = (0.9, 0.95)$, weight decay = 0.1, cosine scheduler, 5% dynamic warmup, gradient clipping = 1.0, BF16, batch=128). Four configurations ($\sim 171$–187M) are compared over 954 steps:

Table 7: Iso-parameter ablation configurations ($\sim$166–187M, 8B tokens FineWeb-Edu).

| Run | Configuration | Params | MLP | Mu | Shared | Experts |
|---|---|---|---|---|---|---|
| run1 | Dense SwiGLU baseline | 171M | Dense (2416) | — | — | 1 |
| run2 | Full Complexity | 187M | Token-Routed (512/e) | ✓ | ✓ | 4 |
| run3 | TR + Shared + Zipf (no Mu) | 187M | Token-Routed (512/e) | — | ✓ | 4 |
| run4 | Mixtral (learned router) | 187M | Mixtral-style (512/e) | — | — | 4 |

Table 8: Full ablation (500M tokens, 954 steps). Average loss measures overall convergence speed.

| Configuration | Params | Avg Loss | $\Delta$ vs Dense |
|---|---|---|---|
| Run 1: Dense (SwiGLU) | 171M | 4.905 | — |
| Run 2: TR + Shared + Mu + Zipf | 187M | **4.793** | $-0.112$ |
| Run 3: TR + Shared + Zipf (no Mu) | 187M | 4.916 | $+0.011$ |
| Run 4: Mixtral (learned router, top-1) | 187M | 4.843 | $-0.062$ |

The full model (Run 2) converges significantly faster than both the dense baseline ($-0.112$) and the Mixtral-style MoE ($-0.050$). Deterministic routing outperforms the learned router in sample efficiency: experts specialize from the first step, without a router learning phase. Without Mu-Guidance (Run 3), Token-Routed is slower than dense ($+0.011$), confirming that the inter-layer equilibrium signal is the key component.

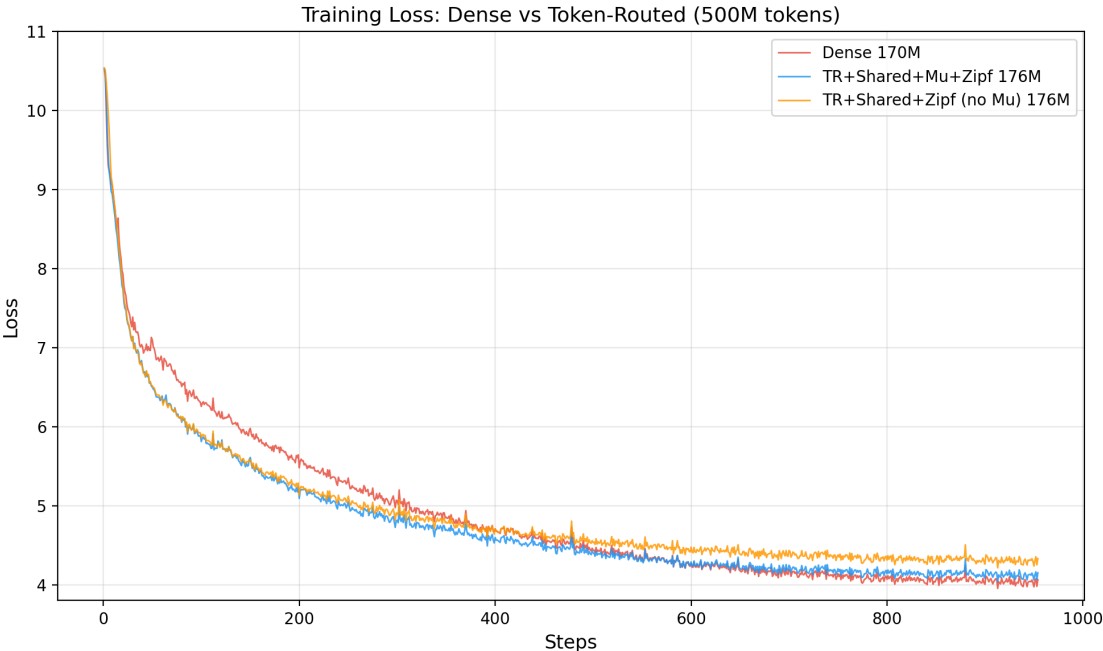

Figure 6: Ablation loss curves (187M, 500M tokens): Dense vs full Token-Routed (Shared+Mu+Zipf) vs Token-Routed without Mu-Guidance. The full model (blue) converges faster throughout.

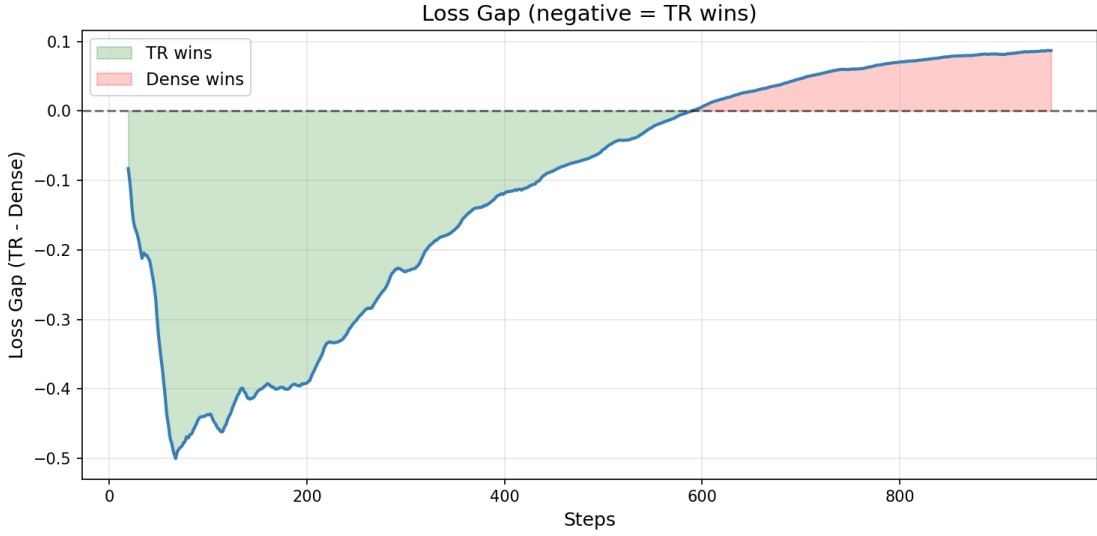

Figure 7: Ablation loss gap TR − Dense (187M, 500M tokens). Negative (green) = TR wins. The TR leads for 95% of training.

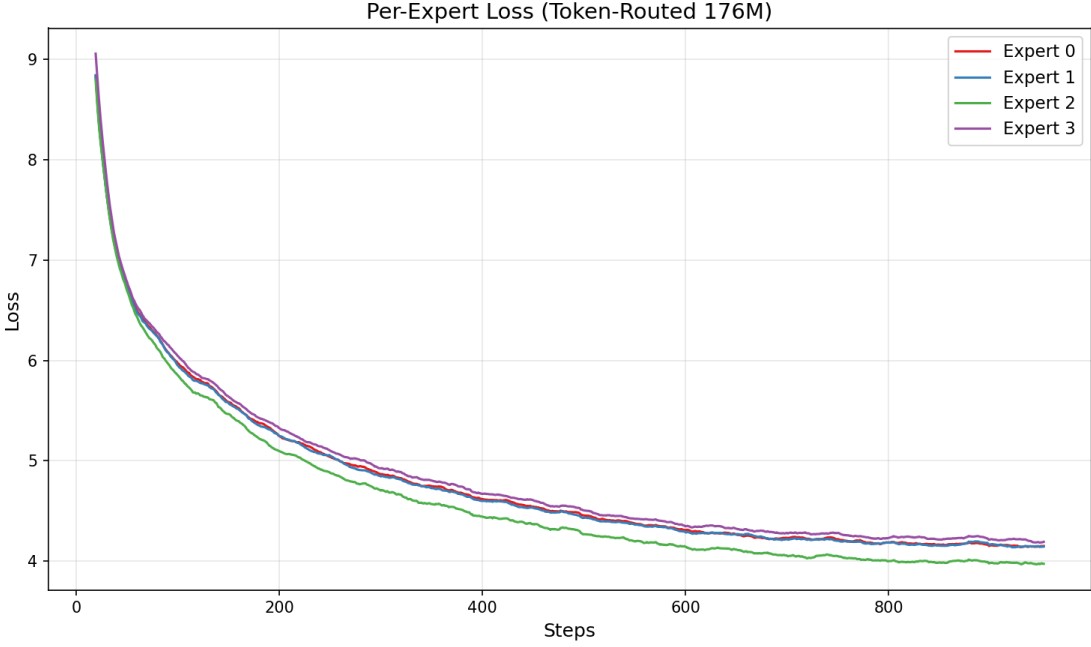

Figure 8: Per-expert loss during ablation training (187M). All 4 experts converge in parallel (max gap 0.23).

## 7.4 Scaling Comparison (300M, 8B tokens)

To validate the corrected architecture at larger scale, we train iso-parameter models (Table 4) on an 8B-token FineWeb-Edu budget. Both runs use the same global batch of 1,048,576 tokens/step (8 GPUs × batch 64 × sequence length 2048), so matched raw step number equals matched tokens seen, and no token-grid interpolation is required.

The two runs start from comparable random-init loss (TR 10.58, dense 10.54) and the Token-Routed model trails the dense baseline during the early phase while its experts are still random and not yet specialized: at step 100 ($\approx$105M tokens) the gap is +0.177 in favor of dense, peaking near +0.31 around step 40. The gap shrinks overall as Zipf-routed experts specialize: the first logged train-loss win appears at step 740 ($\approx$776M tokens), and the first matched validation win appears at step 750. The lead peaks near $-0.023$ around step 2000, then narrows slightly as both runs approach convergence: at step 7620 ($\approx$7.99B tokens, end of budget) Token-Routed reaches train loss **2.9231** versus **2.9324** for dense, and the smoothed final gap over the last 50 logged steps is $-0.0163$ in favor of Token-Routed. Expert utilization remains tightly balanced throughout (0.2481/0.2635/0.2481/0.2403, zero dead experts), confirming that the win is not driven by expert collapse or routing imbalance.

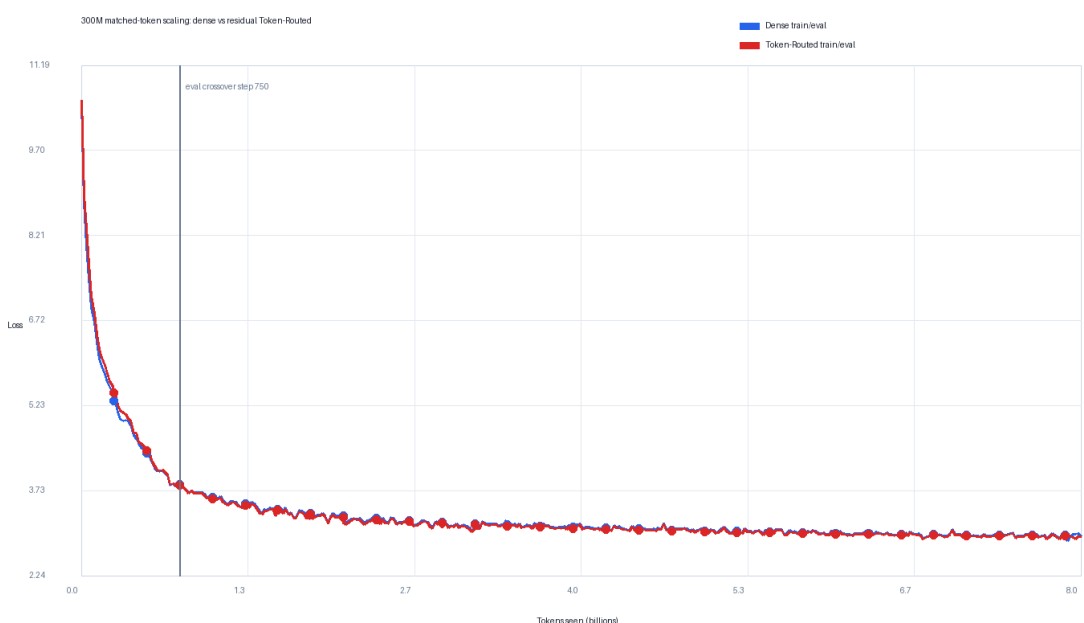

Figure 9: Corrected 300M scaling curves at iso-batch (1,048,576 tokens/step). Token-Routed (red) trails dense (blue) during the first ≈700 steps while experts specialize, reaches parity, then crosses over: by step 1000 (≈1.05B tokens) Token-Routed leads.

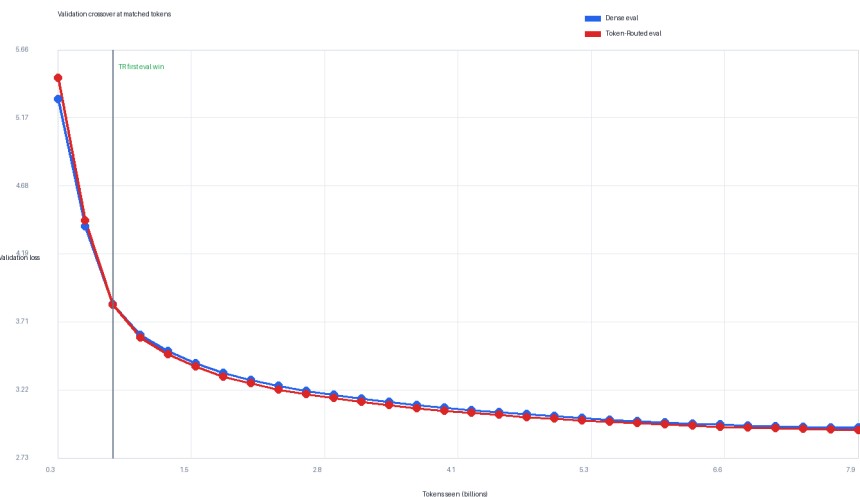

Figure 10: Matched validation-loss comparison. Token-Routed pays an early specialization cost at the first two validation points, then crosses the dense baseline at step 750 (≈786M tokens) and remains ahead through the 8B-token budget.

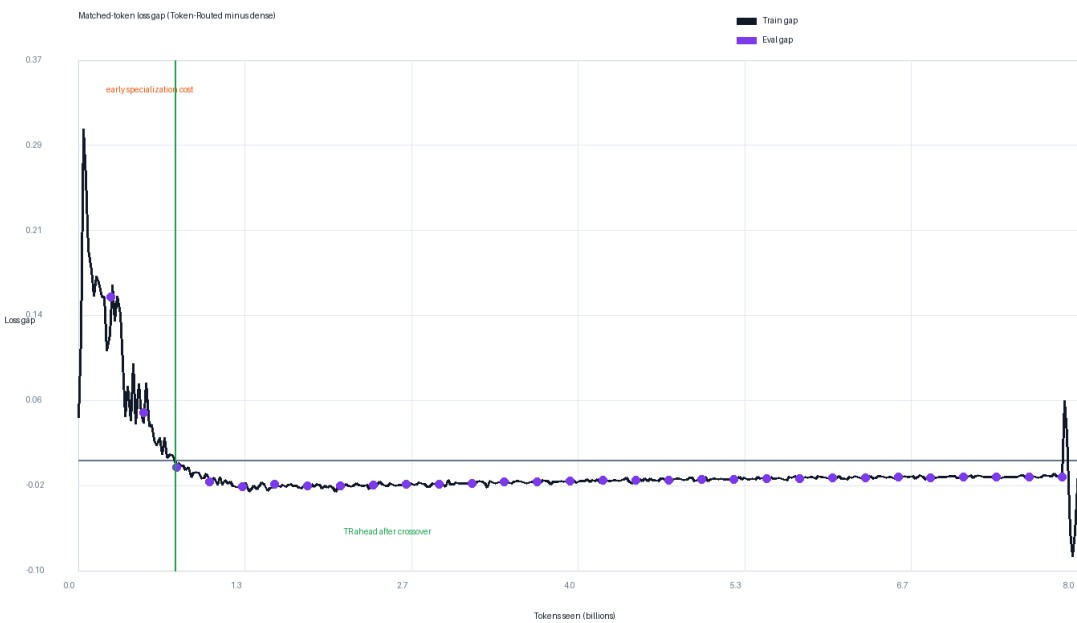

Figure 11: Training-loss gap at matched step (= matched tokens, iso-batch), computed as Token-Routed loss minus dense loss. Negative values indicate Token-Routed ahead. The gap is positive (TR behind) during the early specialization phase and first turns negative at the logged step 740.

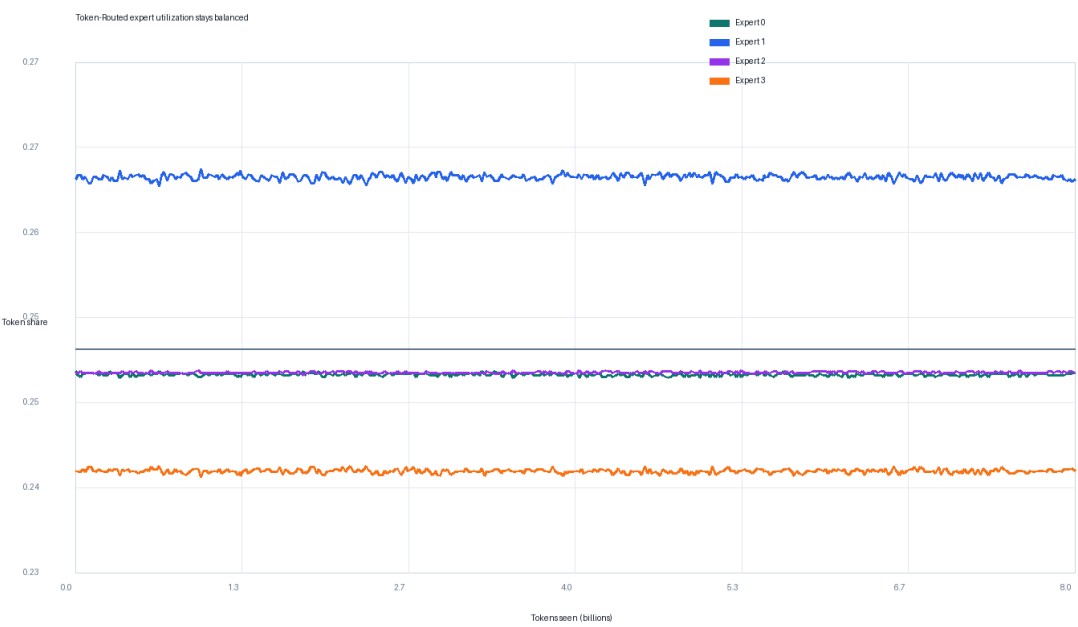

Figure 12: Expert utilization during the corrected 300M Token-Routed run. All four experts remain active and near-balanced; the observed matched-budget win is not caused by expert collapse.

Table 9: 300M scaling comparison at iso-batch (1,048,576 tokens/step), full 8B-token run. Train loss is reported at matched step. Negative gap indicates Token-Routed ahead. Last row is the smoothed mean of the final 50 logged steps.

| Step | Tokens seen | Dense (306.5M) | TR (306.5M, k=2) | Gap (TR − Dense) |
|---|---|---|---|---|
| 100 | 105M | 6.6698 | 6.8464 | +0.177 |
| 500 | 524M | 4.4450 | 4.4796 | +0.035 |
| 700 | 734M | 3.8567 | 3.8619 | +0.005 |
| 1000 | 1.05B | 3.5500 | **3.5324** | **−0.018** |
| 2000 | 2.10B | 3.1953 | **3.1720** | **−0.023** |
| 4000 | 4.19B | 3.0925 | **3.0738** | **−0.019** |
| 6000 | 6.29B | 3.0061 | **2.9906** | **−0.015** |
| 7620 | 7.99B | 2.9324 | **2.9231** | **−0.009** |
| last 50 (smoothed) | — | 2.9446 | **2.9283** | **−0.016** |

**Training throughput.** The Token-Routed model's active compute per token is $2 \times d_{ff}/n$ (one routed expert plus shared expert) vs $d_{ff}$ for dense. Both runs were trained on 8×B300, dense reaching approximately 0.95M tokens/s and Token-Routed approximately 0.75M tokens/s at identical global batch (1,048,576 tokens/step). All quality comparisons above are at matched tokens seen; wall-clock cost per step is comparable since the active routed compute is small relative to the shared expert. The routing table lookup itself is $O(1)$ with zero GPU synchronization.

### 7.5 Inference Performance

The Run 2 model (187M) deployed on vLLM 0.18 with PagedAttention and CUDA graphs achieves a sustained throughput of **8,078 tokens/s** (peak 10,179 tokens/s) serving 100 concurrent requests on a single NVIDIA RTX PRO 6000 GPU (96 GB), with a median time-to-first-token (TTFT) of 29.3 ms and a median inter-token latency (ITL) of 7.9 ms (Figure 13). The deterministic per-token routing (no learned router) is natively compatible with CUDA graph capture, eliminating the CPU–GPU synchronizations required by classical MoE architectures. This property is independent of model size: unlike learned-router MoE which requires all-to-all inter-GPU communication, deterministic routing requires no inter-GPU coordination, suggesting favorable throughput scaling with both GPU count and model size.

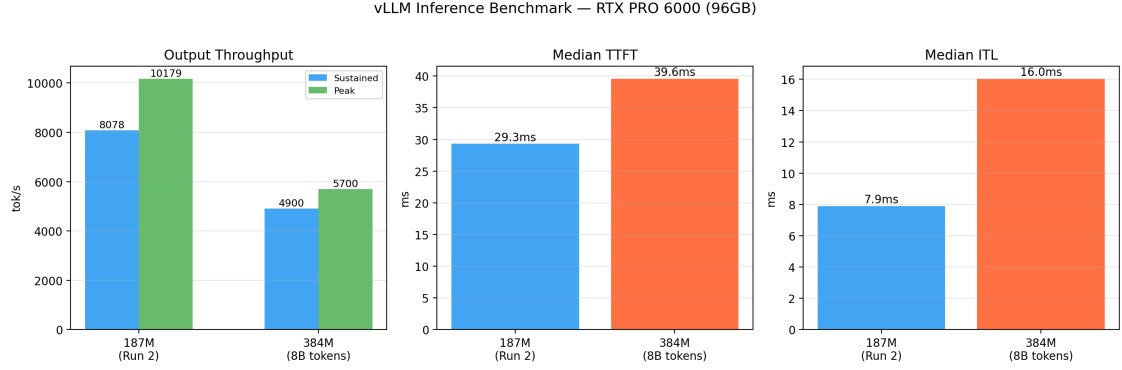

Figure 13: Inference benchmark on a single NVIDIA RTX PRO 6000 (96 GB). 100 requests at 10 RPS, 128 input tokens, 1,900 output tokens. Sustained throughput: 8,078 tok/s; peak: 10,179 tok/s; median TTFT: 29.3 ms; median ITL: 7.9 ms.

### 7.6 Limitations and Future Work

- **No-Mu ablation**: *completed* (Run 3, Table 8). Without Mu-Guidance, Token-Routed is slower than dense (+0.011 in average loss), confirming the essential role of the inter-layer equilibrium signal.

- **Full-run confirmation**: The corrected 300M run loses the first two matched validation points (steps 250 and 500), crosses over at step 750, and remains ahead through the final matched validation point at step 7500. Downstream benchmarks should be rerun from the corrected 300M checkpoints.

- **Standardized benchmarks**: Zero-shot evaluation on ARC-Easy, HellaSwag, and MMLU via `lm-evaluation-harness` should be rerun on the corrected 300M checkpoints.

## 8 Conclusion

We presented COMPLEXITY-DEEP, a language model architecture with three main contributions: (1) Token-Routed MLP with Zipf-balanced greedy bin-packing and Shared Lexical Expert for deterministic assignment without auxiliary losses, (2) Mu-Guided Attention with learnable $\mu_{\text{init}}$ and $\mu$ production after the MLP for expert-aware inter-layer communication, and (3) a training recipe with dynamic warmup (5%) and GPT-style initialization ($1/\sqrt{2L}$).

Component ablation at 187M (500M tokens) shows Token-Routed outperforming the dense baseline in average loss (4.793 vs 4.905, $\Delta = -0.112$). In the corrected iso-parameter 300M scaling run at iso-batch (1,048,576 tokens/step, full 8B-token budget), Token-Routed pays a transient specialization cost during the first $\approx 700$ steps (peak gap +0.31 near step 40), first wins on logged train loss at step 740 and on validation loss at step 750, then stays ahead until the end: the smoothed final train-loss gap over the last 50 logged steps is $-0.0163$ in favor of Token-Routed (2.9283 vs 2.9446). This reverses the earlier scaling conclusion and indicates that the corrected Token-Routed implementation becomes sample-efficient once Zipf-routed experts specialize. Inference on vLLM 0.18 reaches 8,078 tok/s for the 187M model, confirming the deployability benefit of deterministic routing; updated 300M inference numbers will be reported from the corrected checkpoint.

### Broader Impact Statement

This work introduces architectural innovations for language models. While these models have broad applications, they also carry risks of misuse. We release the model weights to enable research while encouraging responsible use.

### Reproducibility Statement

To facilitate reproducibility and review, we provide the complete model architecture implementation as supplementary material (`supplementary_code.zip`). The code (PyTorch 2.0+) will be made publicly available upon acceptance.

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

## A   Training Curves

Additional training curves and analysis figures are available in the supplementary material.

