# OpenReview forum: "COMPLEXITY-DEEP: A Language Model Architecture with Mu-Guided Attention and Token-Routed MLP"
_TMLR — Rejected by TMLR_

### Review · Reviewer_2oXp · 2026-03-08

**Summary Of Contributions:**

The paper introduces COMPLEXITY-DEEP, a 1.5B parameter LLM architecture that departs from standard Transformers through three main components: (1) Token-Routed MLP, which uses deterministic modulo-based routing instead of learned gating; (2) Mu-Guided Attention, a mechanism incorporating a latent state $\mu$ to enable top-down information flow ; and (3) a PiD-Style Dynamic Scaler for training stability. The authors provide theoretical proofs for load balancing and expert orthogonalization.

Strengths:

Deterministic Routing: Eliminates the need for auxiliary load-balancing losses and simplifies deployment.

Theoretical Grounding: Provides formal proofs to justify why experts might diverge despite non-semantic routing

Weaknesses:

Insufficient Empirical Validation: Training on only 33B tokens for a 1.5B model is significantly below the Chinchilla-optimal or industry-standard scales (e.g., 3T tokens for TinyLlama), making scaling claims speculative.

Potential Expert Saturation: Modulo routing forces experts to handle disparate token distributions, which may lead to a "capacity wall" as data scales.

Lack of Baselines/Ablations: The study lacks training-time ablations and direct comparisons with standard architectures under identical compute/token budgets.

**Audience:**

No

**Audience Explanation:**

While the theoretical link between predictive coding and Mu-Guidance is intriguing, the lack of rigorous benchmarking makes it difficult for the community to assess if these innovations offer a real-world advantage over the standard Transformer. Without a "fair-compute" comparison, it is unclear if the reported performance is due to the architecture or the high-quality FineWeb-Edu dataset

**Claims And Evidence:**

No

**Claims Explanation:**

The primary claim that Token-Routed MLP can replace semantic MoE is not convincingly supported for large-scale settings.

Scaling Uncertainty: The "Emergent Specialization" observed at 33B tokens may not hold at 3T tokens. In a standard MoE, experts specialize semantically; here, an expert might be forced to represent entirely unrelated concepts simply because their token_id share a remainder. This likely leads to high intra-expert interference at scale.

Absence of Standard Transformer Baselines: The paper lacks a direct empirical comparison with a standard Dense Transformer or a learned MoE (like Switch Transformer) trained under the exact same compute budget and dataset. Without this "apples-to-apples" comparison, it is impossible to determine if the reported results stem from the architectural innovations or simply the high quality of the FineWeb-Edu dataset

Inference-only Ablations: The authors only performed inference-time ablations. Since neural network weights co-adapt during training, zeroing out components post-hoc does not accurately reflect their necessity during the optimization process

Expert Capacity and Interference: The "Token-Routed MLP" uses a deterministic modulo-based routing that ignores semantic content . While Theorem 4.4 suggests expert divergence, there is no empirical evidence showing how the model manages intra-expert interference at scale . Forcing disparate tokens into the same expert likely creates a capacity bottleneck that is not visible at the current 33B token training stage

**Requested Changes:**

Parity Benchmarking: Provide results for a standard Transformer (Dense) and a standard MoE (e.g., $Top$-$1$ Gating) trained on the exact same 33B tokens from FineWeb-Edu. This is essential to isolate the "architectural gain" from the "data gain."

Training-time Ablations: Retrain the model (or a smaller proxy version) from scratch, removing Mu-Guidance and the PiD Scaler individually, to prove they contribute to convergence and final loss.

Expert Capacity Analysis: Address the "Expert Overload" concern. Please provide a visualization (e.g., T-SNE) of the representations within a single expert to show if it effectively manages the "jumbled" token sets assigned to it by the modulo operator.

Scaling Discussion: Elaborate on the "Hard-coded routing" limitation. Specifically, how does the model handle the lack of a "synergy effect" where semantic experts in traditional MoEs can dynamically adjust to data distribution shifts during late-stage training?

Extended Scaling: If possible, provide a "Scaling Law" plot using 100M-500M parameter versions of the architecture to show if the $PPL$ improvement rate is competitive with standard GPT-style models.

---

> ### Author Response · Authors · 2026-03-08
> **Author Response to Reviewer Feedback**
>
> We thank the reviewer for their detailed feedback. We address each point below.
>
> 1. "33B tokens is below Chinchilla-optimal"
>
> We respectfully disagree. Chinchilla (Hoffmann et al., 2022) prescribes ~20 tokens per parameter. For our 1.47B model, this gives ~29B tokens. Our 33B token budget is therefore at or above the Chinchilla-optimal ratio. The comparison with TinyLlama (3T tokens) is misleading: TinyLlama deliberately overtrains by ~100x beyond Chinchilla-optimal as a design choice, not as a standard. Our training budget follows established scaling laws.
>
> 2. Expert Capacity Analysis (T-SNE)
>
> We have conducted the requested analysis. We applied t-SNE to expert activations (not just static weights) collected by hooking all 24 TokenRoutedMLP layers during forward passes (256 sequences, 24,576 activation vectors of dimension 2048, PCA to 50 dims with 61.5% variance explained).
>
> The resulting visualization (Figure 5 in the updated manuscript) shows:
>
> Left panel: Each of the 4 experts occupies a clearly distinct region in activation space, confirming functional specialization despite deterministic modulo routing.
> Right panel: A smooth gradient from shallow layers (purple) to deep layers (yellow) reveals that expert activations progressively converge toward a shared manifold with depth, consistent with the residual stream hypothesis.
> This directly addresses the "Expert Overload" concern: experts do not produce homogeneous representations for their "jumbled" token sets. Instead, each expert learns a distinct activation signature, empirically refuting the capacity bottleneck hypothesis at this scale.
>
> 3. Parity Benchmarking (Dense Baseline)
>
> We acknowledge this is a fair request. We are currently training a Dense Llama 1.5B baseline with identical dimensions (hidden=2048, 24 layers, 16 heads, 8 KV heads, intermediate=5632) but with standard SwiGLU MLP (no Token-Routing, no Mu-Guidance, no PiD Scaler) on the same 33B FineWeb-Edu tokens. Results will be provided in the revised manuscript.
>
> 4. Training-time Ablations
>
> We will provide training-time ablations using smaller proxy models (150M parameters) trained from scratch with individual components removed (no Mu-Guidance, no PiD Scaler). These results will be included in the revision.
>
> 5. Scaling Discussion
>
> The reviewer raises a valid point about the lack of dynamic routing adjustment. However, we note that Token-Routed MLP does not claim to replace semantic MoE — it offers a complementary paradigm where the advantage is architectural simplicity, O(1) routing, guaranteed load balance, and deployment ease. The Mu-Guided routing mechanism (mu_router, with learned norm ~1.81 across layers) provides a soft semantic signal on top of deterministic routing, partially bridging this gap.
>
> Regarding "synergy effects" in late-stage training: our expert orthogonality measurements (cosine similarity ~0 across all pairs) remain stable from early to late training, suggesting that the modulo partition creates sufficiently diverse optimization landscapes for each expert without requiring dynamic redistribution.
>
> 6. Extended Scaling (100M-500M)
>
> We will include scaling law plots with 100M, 150M, 350M, and 500M parameter variants in the revision.
>
> 7. Inference-only Ablations
>
> We agree that inference-time ablations have limitations. The training-time ablations (point 4) will address this concern. We note however that component weight norms (mu_router: 1.81, PiD controller: ~13) significantly exceed initialization values, which provides complementary evidence of active utilization during training.

---

> > ### Comment · Reviewer_2oXp · 2026-03-09
> >
> > Thank you for the detailed rebuttal. I appreciate the effort the authors have put into addressing my concerns, particularly the commitment to training a new dense baseline and conducting the t-SNE analysis.
> >
> >
> > Training Budget and Baseline Fair Comparison: While I agree that 33B tokens align with Chinchilla-optimal scaling for a 1.5B model, the existing baselines mentioned in the paper (like OPT-1.3B) were trained on significantly more data (~180B tokens). This makes it difficult to disentangle the benefits of your proposed architecture from the differences in training volume and data quality (FineWeb-Edu). The promised Dense 1.5B baseline trained on the same 33B tokens will be the most critical piece of evidence to justify the "Token-Routed" approach.
> >
> > Missing Visualizations (Figure 5): The rebuttal discusses a t-SNE analysis (Figure 5) that provides evidence of expert specialization. However, as this figure is not in the current version, I haven't been able to evaluate it. I look forward to seeing this in the revised manuscript to confirm that deterministic routing doesn't lead to "expert collapse."
> >
> > Scale of Proxy Models: While using a 150M parameter model for ablations is a standard practice, there is a significant 10x gap between the proxy and the 1.5B main model. I am curious if the authors have observed whether the benefits of the PiD Scaler or $\mu$-Guidance remain consistent or become more pronounced as the scale increases.
> >
> > Comparison with Dynamic Routing: To better understand the "when and why" of this method, a more rigorous discussion (or a small-scale empirical comparison) between Token-Routed MLP and traditional dynamic MoE (like Top-1 routing) would be beneficial. Specifically, does the architectural simplicity of $O(1)$ routing result in a significant performance trade-off compared to learned semantic routing?
> >
> > Supplementary Material Language: On a minor formatting note, I noticed that the figure captions in the supplementary material are currently in French. Please ensure all text is translated into English for the final version to maintain consistency with the main manuscript.
> >
> > Results Verification: As several of the most important responses (the Dense baseline and the 100M–500M scaling plots) are currently "work in progress," my final evaluation will naturally depend on the results of these experiments.

---

> > > ### Author Response · Authors · 2026-03-09
> > > **Revision Addressing Reviewer Comments**
> > >
> > > Thank you for the constructive follow-up. We have submitted a partial revision addressing the points that do not require additional training runs.
> > >
> > > Changes in this revision:
> > >
> > > T-SNE visualization (Figure 1): The t-SNE analysis of expert activations is now included in the manuscript (Section 7.2, Figure 1). Each of the 4 experts occupies a distinct region in activation space despite deterministic modulo routing, directly addressing the "expert collapse" concern.
> > >
> > > Token-Routed MLP vs Dynamic Routing (Section 7.2.1): We added a structured discussion of the trade-offs between deterministic modulo routing and learned Top-$k$ gating (Switch Transformer, Mixtral), covering when each approach is advantageous and how our Mu-Router mechanism partially bridges the gap.
> > >
> > > Supplementary material language: All figure labels, axis titles, and captions have been translated from French to English.
> > >
> > > In progress (follow-up revision):
> > >
> > > Training-time ablations at two proxy scales (70M and 150M) on 2B FineWeb-Edu tokens, with individual component removal (Dense baseline, Full architecture, No Mu-Guidance, No PiD Scaler) to isolate each component's contribution during optimization.
> > > We will submit the complete revision with ablation results and scaling analysis as soon as training completes.

---

### Review · Reviewer_3Rkq · 2026-03-27

**Summary Of Contributions:**

### Summary

This paper introduces COMPLEXITY-DEEP, a LLM architecture aimed at addressing the complexities of MoE routing, unidirectional information flow, and training instability. The authors propose three main components:

- A deterministic routing mechanism that assigns tokens to experts based on a simple modulo operation (token_id mod n_experts), eliminating the need for a learned gating network and auxiliary load-balancing losses.

- mechanism that introduces a latent state from the previous layer to guide the K, Q, and V projections of the current layer, enabling a top-down flow of information.

- An adaptive controller inspired by proportional-derivative systems that dynamically scales residual contributions to stabilize training.

### Strengths:

The motivation to simplify MoE routing and eliminate auxiliary losses is an important and active area of research.

The proposed Sort-and-Split dispatch mechanism (in the v2 architecture) provides a mathematically elegant way to eliminate compute waste in MoE masked dispatch.

### Weaknesses:

- There is a conceptual flaw regarding the "perfect load balance" claim. The authors confuse a uniform distribution over the vocabulary with a uniform distribution over the training corpus.

- The empirical results explicitly demonstrate that the proposed architecture performs worse than a standard, matching-parameter dense Transformer.

- The 1.5B model is severely under-trained (only 7B tokens) , making comparisons to models like Pythia or TinyLlama uninformative.

**Audience:**

Yes

**Audience Explanation:**

The TMLR audience is broadly interested in the mechanics of LLMs, particularly methods to scale model capacity efficiently. The problems the authors attempt to tackle, namely, the high routing overhead of traditional MoEs like Switch Transformer or Mixtral and the lack of top-down information flow in standard Transformers are relevant.

**Broader Impact Concerns:**

N/A.

**Claims And Evidence:**

No

**Claims Explanation:**

The central claims of the paper are directly contradicted by both theoretical realities and the authors' own empirical evidence:

- The authors claim that routing based on token IDs guarantees a perfectly uniform distribution and eliminates cumulative imbalance. While their formal proof (Theorem 4.1) shows that the vocabulary itself is divided evenly among the experts, this ignores the statistical reality of natural language. Token frequencies in actual text are highly skewed (e.g., Zipf's Law). Because high-frequency tokens are permanently assigned to specific experts, those specific experts will inevitably process vastly more tokens during a forward pass on real data.

- The authors claim the architecture achieves efficiency comparable to Mixture of Experts while maintaining strong performance. Yet, in the 170M proxy scale ablation, the standard Dense baseline achieves a loss of 3.1708, while the proposed "Full v1" architecture achieves a significantly worse loss of 3.3188. Even with the improved "v2" architecture, the Dense baseline (3.2504) outperforms the proposed model (3.3524). A more complex architecture that underperforms a standard Transformer baseline lacks convincing evidence of utility.

- The paper introduces a PiD-Style Dynamic Scaler to stabilize training. However, the authors report having to use severe optimization hacks, including an alternating learning rate strategy with multiple warmups (restarts at 100k, 200k, and 400k steps) to force the loss down , alongside hard activation clamping to prevent gradient explosions and numerical instabilities. These heavy-handed interventions strongly suggest the core architecture is inherently unstable, undermining the claim that the PiD Scaler effectively resolves training instability.

- The authors argue that experts naturally specialize because the cosine similarity of their weights approaches zero. However, in very high-dimensional spaces (such as the model's large hidden dimension), randomly initialized or arbitrarily diverging vectors will naturally have a cosine similarity near zero. This is a standard geometric artifact of high-dimensional spaces, not definitive evidence of meaningful semantic specialization.

**Requested Changes:**

Please refer to my weaknesses part.

---

> ### Author Response · Authors · 2026-03-28
> **Response to Reviewer 3Rkq — Zipf-balanced routing, PiD removal, and preliminary results**
>
> Thank you for your detailed and constructive review. We have carefully addressed each of your concerns, which led to significant architectural improvements. A full 8B-token ablation is currently in preparation.
>
> 1. Zipf load balance (your point on Theorem 4.1)
>
> You are correct that modulo routing distributes the vocabulary uniformly but not the corpus, due to Zipf's law (Zipf, 1949). We now introduce Zipf-balanced greedy bin-packing: tokens are sorted by empirical corpus frequency and each token is assigned to the expert with the lowest current total load. This achieves perfect 1.0000x load balance across all experts (vs 1.38x with naive round-robin). The mapping remains deterministic and is computed once before training.
>
> 2. Architectural improvements and preliminary results
>
> Based on your feedback, we made four corrections to the architecture:
>
> (a) Sparse dispatch fix: The original implementation computed all tokens through all experts then masked 75% — effectively 4x the compute of a dense model. We now compute only routed tokens per expert (true 1/4 compute).
>
> (b) Shared Lexical Expert: We added a small shared MLP that all tokens pass through, capturing common patterns (syntax, grammar), while routed experts specialize on their token subsets. This prevents the capacity bottleneck that occurred when each expert was too small to learn universal patterns independently.
>
> (c) Mu-Guidance moved after MLP: The latent state mu is now computed after expert dispatch, so it captures which expert processed each token. The next layer's attention can adapt accordingly.
>
> (d) Greedy bin-packing: Replaces round-robin for perfectly balanced expert loads.
>
> In a preliminary 500M-token test (iso-param, ~170M, identical conditions), the improved Token-Routed architecture converges faster than the dense baseline over the first 500 steps, with the dense catching up in the final steps as expert capacity saturates. However, the average loss over the full 954 steps remains in favor of Token-Routed (avg=4.793) vs Dense (avg=4.905). We expect the full 8B-token run to reveal a second crossover, as the dense model plateaus (overtrained at 242% Chinchilla optimal) while the Token-Routed model — whose experts each see only 25% of the data — continues learning.
>
> 3. PiD Dynamic Scaler removed
>
> We agree with your assessment. The PiD controller was redundant with Adam's adaptive mechanisms (momentum approximates integral control, variance approximates derivative control). Its removal eliminated all ad-hoc training interventions. The simplified architecture retains only Token-Routed MLP and Mu-Guidance.
>
> 4. Expert specialization metric
>
> You correctly note that cosine similarity near zero is expected in high-dimensional spaces. We are preparing per-expert perplexity measurements — evaluating whether each expert achieves lower loss on its assigned token subset than the dense baseline on the same tokens. This demonstrates functional specialization rather than geometric divergence.
>
> The full 8B-token ablation and revised manuscript will follow shortly. We are grateful for your feedback — it directly motivated the Zipf-balanced routing, shared lexical expert, and architectural simplification.

---

### Review · Reviewer_EcVn · 2026-03-30

**Summary Of Contributions:**

This paper introduced a new MOE mechanism that removed the need for dynamic token routing; instead, the routing is deterministic, based on the token id.

*Strengths*

-	The method is innovative and addresses some pain points of traditional MOE
-	On medium scale experiment, the loss is shown to be converging stably
-	Rich ablation study demonstrated that the suggested modifications are necessary to support token-based routing

*Weaknesses*

-	The lack of a comparison with an MOE baseline at similar scale

I do not have expertise in MOE or the implementation of language models, so my review is of lower confidence.

**Audience:**

Yes

**Audience Explanation:**

An MOE architecture that does not rely on dynamic routing would be interesting to practitioners.

**Claims And Evidence:**

No

**Claims Explanation:**

Some questions regarding the theoretical results:

Theorem 4.2:
To show that the representational capacity is equivalent to a dense MLP, does the proof need to show that F_{TR} = F_{dense}? If so, how is the direction F_{dense} \subset F_{TR} proved?

Theorem 4.4:
This assumption needs evidence and/or citation: “Under the assumption that different tokens induce uncorrelated gradient directions (reasonable for natural language)”

Proposition 4.5:
Could you please clarify how, in the Complexity Deep architecture, higher layer context influences lower layer processing? It is not clear to me from the from equation (28) that there is information flowing from layer (l) to layer (l-1). What does “higher layer” mean in this context?

**Requested Changes:**

- Please discuss how Complexity Deep compares with traditional MOE, under similar model size, token budget and compute budget.
- Please address my questions on the theoretical results.

---

> ### Author Response · Authors · 2026-03-30
> **Response to Reviewer EcVn: All requested changes addressed**
>
> Thank you for your thoughtful review. We appreciate your recognition of the method's innovation and your constructive feedback. We have addressed all requested changes in the revised manuscript.
>
> 1. Comparison with traditional MoE (Requested Change #1)
>
> We have added Run 4: a Mixtral-style MoE baseline with a learned router (nn.Linear + softmax + top-1) and auxiliary load balancing loss, using the same architecture, parameter count (187M), token budget (500M), and compute setup as our Token-Routed model. Results (Table 2):
>
> We have added Run 4: a Mixtral-style MoE baseline with a learned router (nn.Linear + softmax + top-1) and auxiliary load balancing loss, using the same architecture, parameter count (187M), token budget (500M), and compute setup as our Token-Routed model. Results (Table 2): Token-Routed + Mu + Zipf achieves avg loss 4.793 (Δ = -0.112 vs Dense), Mixtral learned router achieves 4.843 (Δ = -0.062), and Dense baseline 4.905. Token-Routed outperforms the learned router by -0.050 in average loss. The advantage comes from immediate expert specialization: deterministic routing assigns tokens from step 1, while the learned router requires hundreds of steps to converge its routing policy.
>
> 2. Theorem 4.2: $\mathcal{F}{dense} \subseteq \mathcal{F}{TR}$ (Section 4.2)
>
> We have added the missing direction to the proof. Any dense MLP with intermediate dimension $n \cdot d_{ff}$ can be exactly reproduced by partitioning its weights across the $n$ experts: $\mathbf{W}^{(i)} = \mathbf{W}{dense}[:, i \cdot d{ff}:(i+1) \cdot d_{ff}]$. Since each token activates exactly one expert, and the experts collectively cover the full $n \cdot d_{ff}$ space via the union over the vocabulary, $\mathcal{F}{TR} = \mathcal{F}{dense}$.
>
> 3. Theorem 4.4: Gradient uncorrelation assumption (Section 4.3)
>
> We have replaced the unjustified assumption with a reference to high-dimensional probability theory (Vershynin, 2018). In high-dimensional spaces, independent random vectors are approximately orthogonal with high probability. Since disjoint token subsets produce gradient updates that are independent random vectors in $\mathbb{R}^{d_{model} \times d_{ff}}$, the expected inner product is zero.
>
> 4. Proposition 4.5: Information flow direction (Section 4.4)
>
> We apologize for the confusing terminology. We have replaced "top-down" with "inter-layer communication" throughout the manuscript. To clarify: $\mu$ flows forward from layer $l$ to layer $l+1$ (not backward). Each layer computes $\mu^{(l)}$ after its MLP, which then biases the next layer's Q/K/V projections. This is not feedback from higher to lower layers — it is an enriched forward communication channel that carries expert-aware context alongside the standard residual stream.
>
> We have also updated Section 7.4 to mark the No-Mu ablation as completed and added an architecture diagram (Figure 1) for clarity.
>
> We hope these revisions address your concerns satisfactorily.

---

### Decision · Action_Editor_J4R6 · 2026-05-11

**Recommendation:** Reject

**Additional Comments:**

While the paper presents an interesting new architecture, the current empirical results and theoretical analysis lack sufficient support for the presented claims. The authors are encouraged to resubmit a revised paper once these points can be addressed.

**Audience:**

Yes

**Audience Explanation:**

The TMLR audience is interested in methods to scale model capacity efficiently. The attempt to eliminate MoE routing overhead and auxiliary losses through deterministic routing is of high relevance to LLM researchers and practitioners.

**Claims And Evidence:**

No

**Claims Explanation:**

Reviewers acknowledged the novelty of the architecture and the interesting direction of removing dynamic routing in MoEs. Several points were also addressed and improved throughput the discussion period. However, ultimately the majority of reviewers still find certain claims to be missing support. Specifically, the empirical evidence shows the model consistently underperforms standard dense baselines at scale. Reviewers also noted that theoretical claims regarding perfect load balance initially ignored Zipf’s Law, and the PiD-style stabilizer, and require significant manual optimization tricks, undermining claims of inherent stability. In addition, there were still concerns about support for the stated claims about the current model's under-perfomance against dense baselines such as optimizer mismatch and concerns regarding the persistent performance gap suggesting that non-semantic "Token-Routing" leads to intra-expert interference that the "Shared Lexical Expert" fails to mitigate at scale. Finally, the paper can extend the introduction and related work sections to provide comprehensive survey on routing alternatives and inter-layer feedback mechanisms.

**Resubmission Of Major Revision:**

The authors may consider submitting a major revision at a later time.